# Quantifying the ecological carrying capacity of grasslands in Inner Mongolia

**Caiyun Guo** [ID][1]ᵒ*, **Shuyu Song**[2]ᵒ, **Dongsheng Zhao**[2], **Shilin Luo**[1]‡, **Lingchun Yang**[3]‡, **Gege Xie**[2]

1 School of Civil Engineering, Changsha College, Changsha, Hunan, China, 2 Key Laboratory of Land Surface Pattern and Simulation, Institute of Geographical Sciences and Natural Resources Research, Chinese Academy of Sciences, Beijing China, 3 School of Physical Education, Changsha College, Changsha, Hunan, China

ᵒ These authors contributed equally to this work.
‡ SL and LY also contributed equally to this work.
* guocy.17b@igsnrr.ac.cn

**Data Availability Statement:** All relevant data are within the paper and its Supporting Information files.

**Funding:** This study was supported by the Scientific Research Foundation of the Hunan Provincial Education Department (21C0744). The

## Abstract

Quantifying the ecological carrying capacity has emerged as a crucial factor for maintaining ecosystem stability for sustainable development in vulnerable eco-regions. Here, we propose a new framework for ecological carrying capacity quantification suitable for vulnerable eco-regions. We applied this framework to calculate the ecological carrying capacity of Inner Mongolia from 1987–2015 and used a geographical detector to identify the driving factors behind spatial heterogeneity. Our results revealed the following. (1) The above-ground net primary production (ANPP) required to support the ecosystem service of soil conservation ($ANPP_{SC}$) decreased from northeast to southwest, whereas the distribution pattern of ANPP required to support the ecosystem service of sand fixation ($ANPP_{SF}$) exhibited a contrary trend. The average annual ANPP required to support the ecosystem service of natural regeneration ($ANPP_{NR}$) in Inner Mongolia from 1987 to 2015 was 101.27 $gCm^{-2}year^{-1}$, revealing a similar spatial distribution with ANPP. (2) The total ecological carrying capacity of Inner Mongolian grassland was 78.52 million sheep unit $hm^{-2}$. The regions with insufficient provisioning service capability accounted for 4.18% of the total area, primarily concentrated in the east and northwest. (3) The average optimal livestock number for grasslands in Inner Mongolia was 1.59 sheep unit $hm^{-2}$ from 1987–2015, ranging from 0.77 to 1.69 sheep unit $hm^{-2}$ across different zones. The average ecological carrying capacity of the cold temperate humid, medium-temperate arid, and warm temperate semi-humid regions was less than 1.08 sheep unit $m^{-2}$, suggesting a need to prohibit grazing in these areas. (4) The primary influencing factors affecting ecological carrying capacity distribution were normalized difference vegetation index (NDVI), precipitation, and soil type. The framework developed herein can help identify sustainable development potential from the ecosystem service perspective and effectively contribute to decision-making in grassland ecosystem management.

funder had no role in study design, data collection and analysis, the decision to publish, or the preparation of the manuscript.

**Competing interests:** The authors have declared that no competing interests exist.

## Introduction

With the pursuit of sustainable development becoming urgent worldwide, the coexistence of societal development and ecological protection is crucial. Assessing the sustainability of vulnerable eco-regions, especially grasslands, is a central issue [1–3]. Grasslands, which cover approximately 40.5% of the Earth's ice-free terrestrial surface and are inhabited by 800 million people [4, 5], face increasing pressure from climate change and intensive land use [6]. Given the limited grassland resource supply, coordinating the utilization of grasslands by natural and human systems is crucial. Exceeding the carrying threshold in grazing intensity for husbandry production activities leads to irreversible damage to the grassland ecosystem, resulting in grassland degradation [7]. Ecological carrying capacity (ECC), a key indicator for measuring regional sustainable development in relation to ecology, society, and economy, provides guidance for grassland management [8, 9]. Therefore, ECC must be ascertained to maintain ecosystem stability in regional development.

Various methods have been used to quantify the ECC variation patterns across different ecosystems, such as the ecological footprint, net primary productivity, pressure-state-response model system model, and human appropriation of net primary production (NPP) methods [10]. The ecological footprint method [11] and net primary productivity method [12] have been widely used to evaluate sustainable development. However, the ecological footprint method ignores landscape dynamics during evaluation. The net primary productivity method has difficulty assessing the threshold of ecosystem sustainability [13]. The precision of the pressure-state-response model evaluation results is proportional to the breadth of the selected indicators [14]. Comprehensive evaluation methods [15] and system models [16] have fully considered the factors affecting ECC from a holistic perspective, whereas many uncertainties arise from model construction, index selection, and parameter calibration. The status of an ecosystem can be evaluated systematically and comprehensively based on a system model; Screening various indicators that can reflect the status of the ecosystem is crucial [17]. Most previous studies have neglected ecosystem processes and functions and the synergies and correlations of environmental and social factors among ecosystem services. A specific framework and systematic evaluation should be established to assess the ECC of vulnerable eco-regions by coupling the interactions between ecosystems and socio-economic systems to maintain ecological balance. Furthermore, identifying the key factors driving ECC variation is important, as it is influenced by the efficiency of socio-economic development and resource, environmental, and ecological supplementation [18]. Ecosystem services proposed by the Millennium Ecosystem Assessment have been defined as the various benefits humans obtain from the ecosystem, including direct and indirect benefits [19]. The supply of ecosystem services is affected by human preferences, cognitive levels, and behavioral decision-making [20]. Increased access to one ecosystem service may lead to decreased access to other types. Previous studies have revealed that trade-offs usually occur between provisioning and regulating services [21], which provide a theoretical basis for ECC research. Assessing ECC by balancing ecosystem services represents a new approach to guide development and conservation [22]. The ecosystem supports the human system by providing ecosystem services, presenting a finite resource base upon which human activities depend [23]. Humans act as interveners in interactive systems where human decisions affect ecosystem resources [24]. Ecosystems are complex systems with the resilience to maintain the robustness of their structure and function; intensive utilization may weaken ecosystem services and degrade future ECC, thereby restricting socio-economic development. In contrast, if the exploitation of ecosystem services is consistently below supply, the ecosystem can support regional development and maintain the ecological balance simultaneously, positively affecting ECC improvement. The assessment of ECC should

focus on ecosystem stability and the appropriate use of ecosystem services without degrading them [25].

The Inner Mongolian grassland is the largest pasture in China, located in arid and semi-arid fragile regions, and constitutes an important ecological barrier in northern China. This region is a production base and the main livestock husbandry industry [26]. It performs vital functions, such as soil and water conservation and biodiversity protection. Due to climate change and continuous overgrazing to meet the increasing human demand, Inner Mongolian grassland ecosystems have been severely degraded, endangering regional ecosystem services [27, 28]. To relieve grazing pressure and restore fragile ecosystems, the Chinese government launched a series of ecological programs that have partially improved grassland ecosystems [29]. Implementing these programs requires funding support, putting a financial burden on the government [30]. Therefore, the ECC of Inner Mongolian grasslands must be evaluated to foster a scientific grassland utilization pattern for ecological conservation and sustainable development.

In this study, we propose a new framework for ECC quantification to ensure ecological function integrity suitable for vulnerable eco-regions. We discuss the value of NPP occupied by key ecosystem services and the NPP required by the ecosystem to maintain ecological integrity in the study area from 1987–2015. In addition, a geographical detector was used to identify the factors driving spatial heterogeneity. This study will supplement existing research in the following ways.

1. Compared with traditional ECC calculations, we estimated the ECC of grasslands in Inner Mongolia under the premise that basic ecosystem services are met. This is the above-ground net primary production (ANPP) allocation process between ecosystem functional maintenance and livestock husbandry. We established a novel research framework for assessing ECC suitable for vulnerable eco-regions based on ecosystem services, which has considerable advantages in guaranteeing all the resources required for ecological functions. This study will provide a useful framework for developing sustainable livestock farming in the Inner Mongolian grassland ecosystem and other regions.

2. We identified the coupling relationships between key variables driving variations in ECC using a geographical detector. We also analyzed the spatial patterns of ECC and identified the driving factors affecting their spatial heterogeneity.

From a regional perspective, Inner Mongolia grassland was selected as the research object based on the largest longitude span and most grassland types in ecologically fragile areas in China. Studying the ECC of Inner Mongolia can help understand the grassland utilization of some pastures in Northwest China and establish early warning mechanisms in the other ecologically vulnerable areas.

## Materials and methods

### Study area

Inner Mongolia, the northern frontier province of China, lies between 37°24′-53°23′N and 97°12′-126°04′E, with a total area of approximately $1.18 \times 10^6$ km². The topography is inclined from southwest to northeast with an average elevation of 1000–1200 m. Inner Mongolia has a temperate zone continental monsoon climate characterized by strong wind and drought, with an average annual temperature of 0–8°C that decrease progressively from south to north and an average annual precipitation of 50–450 mm that declines along a gradient from east to west. The region is divided into six unique eco-regions based on differences in hydrothermal

conditions, topography, and other factors. Grasslands are the major vegetation type in Inner Mongolia, accounting for approximately 67% of the total area [31], including desert steppe, typical steppe, and meadow steppe. They play a major role in animal husbandry (AH) and ecological conservation. Due to climate change and human activities, severe soil erosion and grassland degradation have become serious problems in this region, greatly harming the sustainable use of grasslands (**Fig 1. The study area and the sampling plots.**).

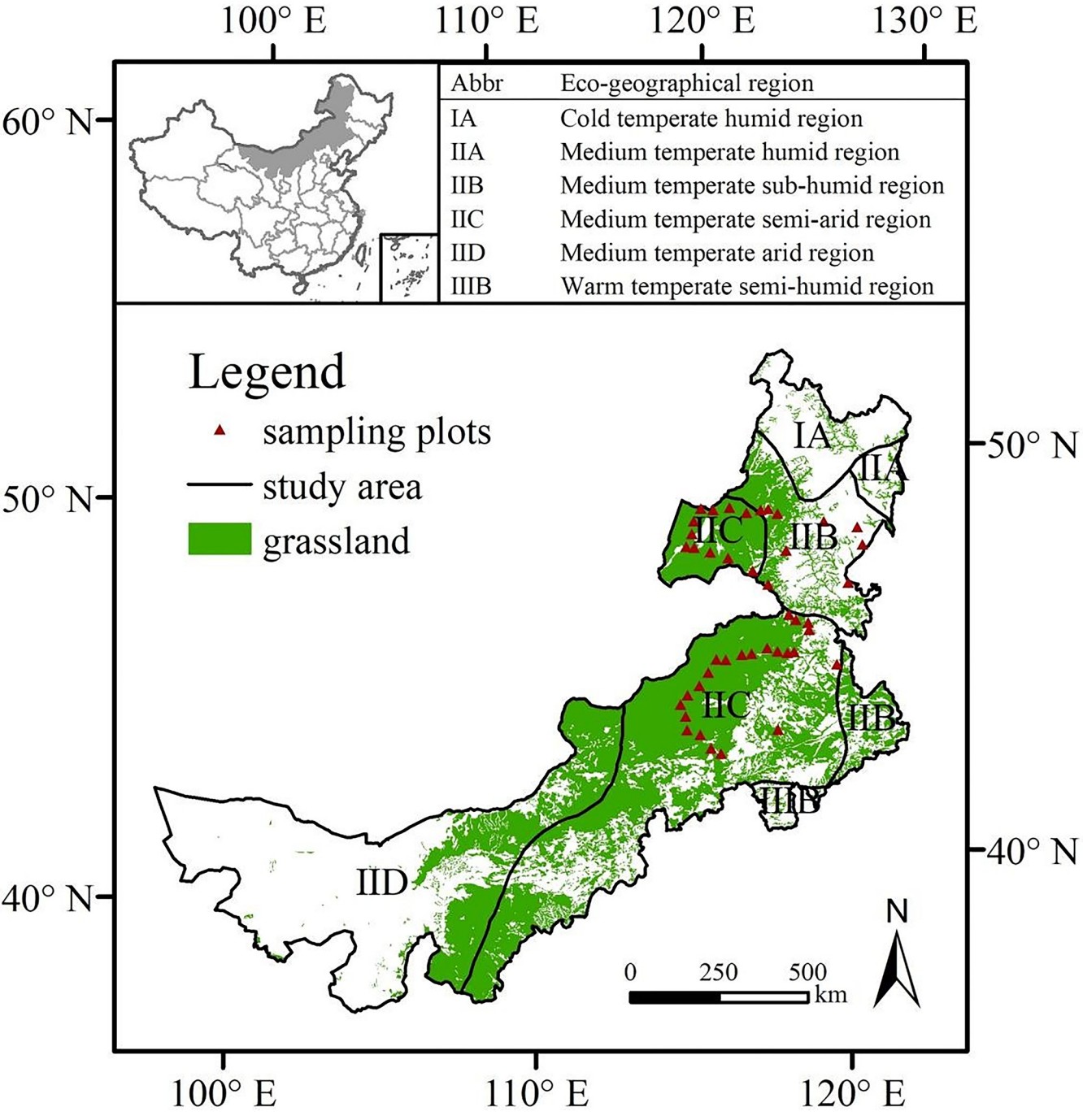

**Fig 1. The study area and the sampling plots.**

## Methods

**Research framework for ECC based on ecosystem services.** Given that the benefits ecosystems provide to humans are limited, trade-offs among various ecosystem services make it challenging to meet multiple needs simultaneously [32]. The premise of ecosystem utilization to generate economic benefits is to ensure a sustainable supply of ecosystem services to maintain natural ecosystem functions. The ECC assessment from the ecosystem services perspective focuses on the limitations of regulating and supporting provisioning services, with the sustainable use of various ecosystem services at its core.

As a major ecological component of grassland ecosystems, vegetation is the primary energy producer for the ecosystem and the basis of provisioning, supporting, and regulating services, which characterize the natural ecological function of a region and its capacity to deliver a set of ecosystem services [33]. NPP is a measure of carbon assimilated in a given period by vegetation and can be used as an indicator of the human appropriation of ecosystem productivity and energy available to other species and the inorganic environment of the ecosystem [34]. To maintain the long-term availability of ecosystem services, part of the vegetation must be retained to avoid negative effects on the ecosystem; therefore, only part of the ANPP can be harvested as provisioning services. Hence, ANPP allocation to natural and human systems is a breakthrough in ECC estimation [35]. The mathematical expression is as follows:

$$\{ANPP\} = \{ANPP_{SC}, ANPP_{SF}, ANPP_{NR}, UNPP, \ldots\} \quad (1)$$

where $ANPP$ is the above-ground NPP ($gCm^{-2}$), $ANPP_{SC}$ is the part of ANPP required to support the ecosystem services of soil conservation (SC), $ANPP_{SF}$ is the part of ANPP required to support the ecosystem services of sand fixation (SF), $ANPP_{NR}$ is the part of ANPP required to support the ecosystem service of natural regeneration (NR), $ANPP_{ES}$ is the maximum ANPP of the key services. $UNPP$ is ANPP that can be used in AH ($gCm^{-2}$).

The entire ANPP of Inner Mongolia can be regarded as a set containing each part of the ANPP required to support ecosystem services. Ecosystems provide various ecosystem services simultaneously; thus, retaining the maximum NPP required by ecosystem services can partially ensure the utilization of other ecosystem services. The mathematical expression is as follows:

$$ANPP_{ES} = MAX\{ANPP_{SC}, ANPP_{SF}, ANPP_{NR}, \ldots\} \quad (2)$$

Considering the characteristics and major environmental problems of Inner Mongolian grasslands, four critical ecosystem services related to regional sustainable development were selected to evaluate ECC: AH, SC, SF, and NR. AH refers to the major food provisioning service; SC and SF are important functions and ecosystem services for preventing wind and water erosion [36]. NR reflects the self-recovery capacity and ecosystem service of grassland, which is the vital function and service of grassland ecosystems. When the grazing intensity exceeds a certain threshold, grassland loses its regeneration capacity [37]. AH and these ecosystem services compete for the occupation of the ANPP. The remaining part of the ANPP after deducting $ANPP_{ES}$ from ANPP is available for AH (UNPP). The ECC is the capability of an ecosystem to supply provisioning services to support livestock husbandry development in the context of maintaining the provision of natural ecosystem services. The mathematical expressions are as follows:

$$UNPP = ANPP - ANPP_{ES} \quad (3)$$

$$ECC = \frac{UNPP \times \alpha}{\beta \times 365} \qquad (4)$$

ECC is the ecological carrying capacity, represented by the livestock carrying capacity (sheep unit m-2); α is the conversion coefficient of standard hay, which is 1 in Inner Mongolian grassland; β α is the daily intake of 1 sheep unit, 1.8 kg.

The grassland ANPP was estimated based on the remote sensing inversion method for grassland biomass proposed by Gong [38] and the ratio of ANPP to above-ground biomass (AGB) [39]. In addition, to facilitate comparison with the actual stocking rate, *UNPP* was converted into standard sheep units and regarded as the ECC of Inner Mongolian grassland (Appendix 1. Supplementary Methods Description in S1 Table).

The Revised Universal Soil Loss Equation [40] and Revised Wind Erosion Equation [41] were used to estimate the minimum vegetation coverage required to reduce soil erosion to below the allowable amount of soil erosion. The allowable amount of water erosion is 5 thm$^{-2}$year$^{-1}$, and the allowable amount of wind erosion is 2 thm$^{-2}$year$^{-1}$ according to the standards for classification and gradation of soil issued by the Ministry of Water Resources of the People's Republic of China (Appendix 1. Supplementary Methods Description in S1 Table).

For *ANPP$_{NR}$*, primary productivity initially increases with grazing intensity and decreases with more intensive grazing intensity [37]. Previous studies have suggested that ANPP is maximized when the utilization rate is 45–51% [42–44]. Therefore, *ANPP$_{NR}$* was specified as 50% of the ANPP.

**Identification of driving factors of ECC.** We used a geographical detector to quantify the influence of the selected eight factors on the spatial distribution of ECC and to identify the main factors that influence the spatial heterogeneity of ECC. This statistical tool was used to explore the determinants of heterogeneity. The assumption is that if a factor contributes to ECC, its spatial distribution will exhibit similarity [45]. The explanatory power of each factor in the ECC was measured using the *q*-statistic:

$$q = 1 - \frac{\sum_{h=1}^{L} N_h \rho_h^2}{N\rho^2} \qquad (5)$$

where $h = 1, 2, \ldots, L$ is the stratification of influencing factor *X*; *L* is the number of strata; $N_h$ and *N* are the numbers of samples in stratum *h* and the entire study area, respectively; $\sigma_h^2$ and $\sigma^2$ are the variance of ECC in stratum *h* and the entire study area, respectively. The value of *q* is within [0, 1]; the larger its value, the stronger the explanatory power of the influencing factor *X*.

The geographical detector can also be applied to investigate whether two factors simultaneously enhance or weaken explanatory power or whether they influence ECC independently. The interaction relationships can be classified into five types by comparing the *q* value of the interaction ($q(X_1 \cap X_2)$) with $q(X_1)$ and $q(X_2)$, as summarized in Table 1.

Using a geographical detector, the individual and interactive influences of multiple factors on the ECC were quantified in various eco-geographical regions of Inner Mongolia. Eight natural factors were considered: temperature, precipitation, wind speed, sunshine duration, elevation, slope, soil type, and NDVI. Based on the data format requirements of the geographical detector, all the independent variables should be discrete. Therefore, the natural break method was used to convert the continuous data into categorical data.

**Data collection and processing.** Meteorological data from 231 meteorological stations within and around Inner Mongolia from 1987–2015 were provided by the National

**Table 1. Types of interaction relationship for the two factors.**

| Description | Interaction type |
|---|---|
| $q(X_1 \cap X_2) < \text{Min}(q(X_1), q(X_2))$ | Weaken, nonlinear |
| $\text{Min}(q(X_1), q(X_2)) < q(X_1 \cap X_2) < \text{Max}(q(X_1), q(X_2))$ | Weaken, univariate |
| $q(X_1 \cap X_2) > \text{Max}(q(X_1), q(X_2))$ | Enhanced, bivariate |
| $q(X_1 \cap X_2) = q(X_1) + q(X_2)$ | Independent |
| $q(X_1 \cap X_2) > q(X_1) + q(X_2)$ | Enhanced, nonlinear |

Meteorological Information Center (http://data.cma.cn), including daily temperature, precipitation, wind speed, and sunshine duration. Missing data (no more than 5%) were interpolated using data from adjacent stations during the same period. All meteorological data were interpolated into raster data at a resolution of 1 km using the kriging interpolation method. Long-term series of daily snow depth datasets for China were obtained from the National Tibetan Plateau Data Center (http://data.tpdc.ac.cn) at a resolution of 25 km.

Soil data with a resolution of 0.5˚ × 0.5˚ were obtained from the soil map based Harmonized World Soil Database provided by the National Tibetan Plateau Data Center. These datasets included clay, silt, sand, organic matter, and calcium carbonate contents.

Land use data with a resolution of 1 km × 1 km and 5-year intervals from 1990 to 2015 were provided by the Resource and Environment Science and Data Center (http://www.resdc.cn). Land use data were interpreted from Landsat TM/ETM and Landsat 8 remote sensing images using the human-computer interactive interpretation method. They were divided into six classes (cropland, woodland, grassland, water body, built-up area, and unused land) and 25 subclasses.

Satellite-based NDVI products with an 8 km × 8 km resolution and 15-day intervals were acquired from the GIMMS NDVI3gV1.0 dataset (https://ecocast.arc.nasa.gov). This dataset is an upgraded product of the GIMMS NDVI 3 g data after reducing the noise interference from volcanic eruptions and sensor-induced errors.

Digital Elevation Model (DEM) data with 90 m × 90 m spatial resolution were derived from SRTM data V4.1 (https://srtm.csi.cgiar.org). The topographical factor tool for the soil erosion model V2.0 was provided by the National Earth System Science Data Center (http://www.geodata.cn).

The AGB data used in this study were obtained from a field survey conducted in Inner Mongolia from July 23 to August 1, 2015, when the grasslands reached their peak biomass. A total of 218 small quadrats (1 m × 1 m) were sampled from 43 plots. The geographical position, grassland type, and plant species composition were recorded within each plot, and the average height and vegetation coverage were measured. All AGB within the field quadrats was harvested at the ground level. The grass samples were returned to the laboratory, dried at 65˚C for 48 h, and weighed to obtain dry weight. The AGB for each sampling plot was obtained by averaging the dry weights of the quadrats.

## Results

### Spatial variation of ANPP$_{SC}$, ANPP$_{SF}$, ANPP$_{NR}$, and ANPP$_{ES}$

The average annual ANPP$_{SC}$ from 1987 to 2015 in Inner Mongolia grassland was 58.67 gCm$^{-2}$year$^{-1}$ (Fig 2A). Being mainly affected by precipitation and topography, ANPP$_{SC}$ exhibited obvious spatial heterogeneity, generally decreasing from the northeast to the southwest. High values of ANPP$_{SC}$ are primarily distributed in the eastern mountainous areas with characteristics of sufficient precipitation and steep topography. The ANPP$_{SC}$ at the eco-geographical

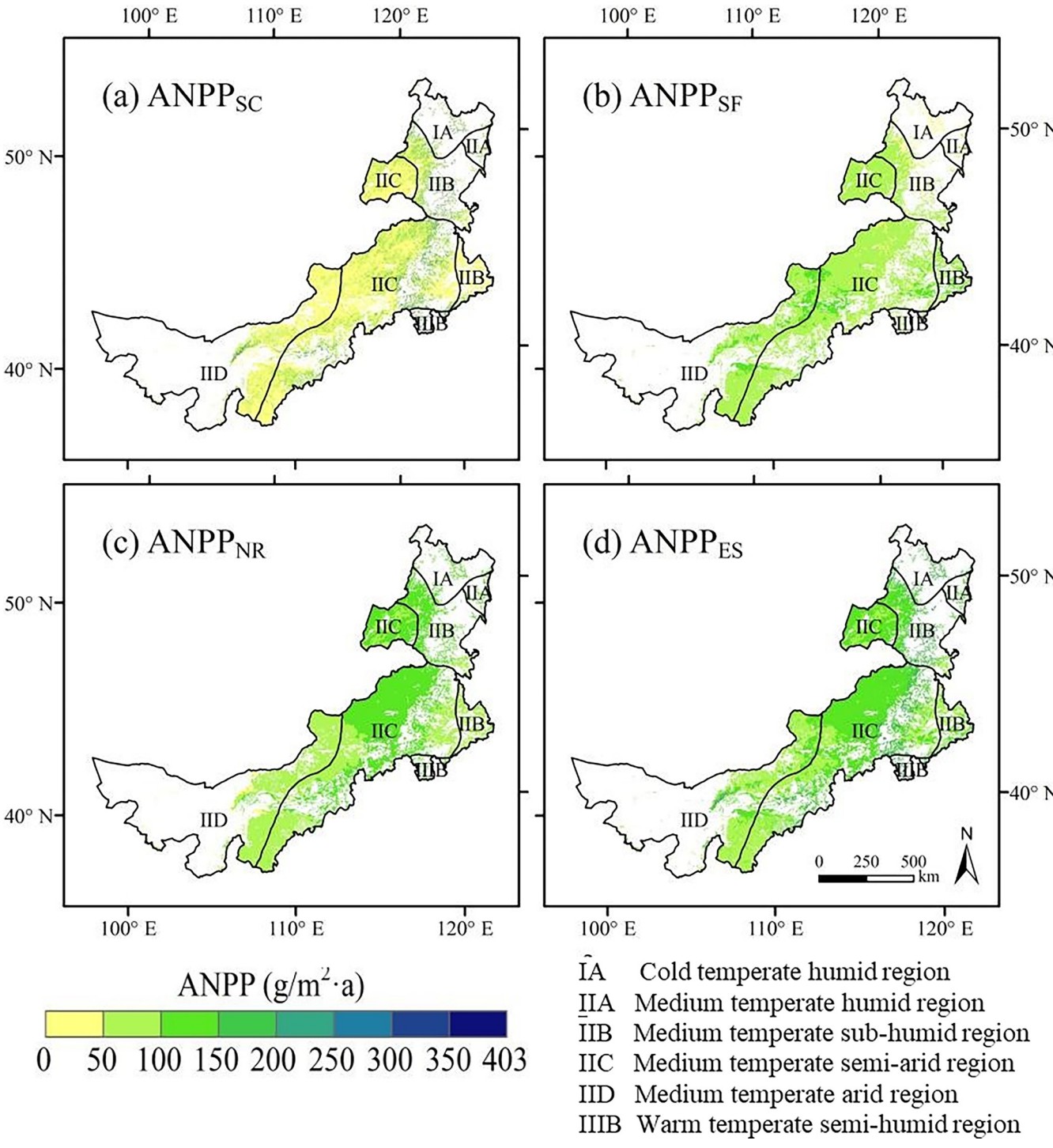

**Fig 2.** Spatial patterns of multi year average (a) $ANPP_{SC}$, (b) $ANPP_{SF}$, (c) $ANPP_{NR}$, (d) $ANPP_{ES}$ in Inner Mongolia grassland from 1987 to 2015.

regional scale revealed spatial distinctions. The average $ANPP_{SC}$ of the warm-temperate semi-humid region was the highest level at 161.45 $gCm^{-2}year^{-1}$. The medium-temperate semi-arid region and the medium-temperate arid region displayed a low level of $ANPP_{SC}$, with the average value of 57.20 and 46.29 $gCm^{-2}year^{-1}$, respectively (**Fig 2. Spatial patterns of multi year**

**average (a) ANPP$_{SC}$, (b) ANPP$_{SF}$, (c) ANPP$_{NR}$, (d) ANPP$_{ES}$ in Inner Mongolia grassland from 1987 to 2015.**).

Over the period 1987–2015, the average annual ANPP$_{SF}$ was 77.53 gCm$^{-2}$year$^{-1}$ (Fig 2B). The distribution pattern of ANPP$_{SF}$ exhibited a trend contrary to that of ANPP$_{SC}$, increasing from the northeast to the southwest with relatively slight changes. The areas with higher ANPP$_{SF}$ were mostly located in the west, where aeolian sandy and brown soils are prone to erosion by strong winds. A comparison between the ANPP$_{SF}$ in different eco-geographical regions revealed low spatial heterogeneity. The average annual ANPP$_{SF}$ of six regions ranged from 42.91 to 89.36 gCm$^{-2}$year$^{-1}$, with the highest value in the medium-temperate arid region and the lowest in the cold temperate humid region.

The average annual ANPP$_{NR}$ in the study area from 1987 to 2015 was 101.27 gCm$^{-2}$year$^{-1}$, revealing a substantial decreasing trend from the northeast to the southwest (Fig 2C). ANPP$_{NR}$ followed a spatial distribution similar to that of ANPP. The highest value of ANPP$_{NR}$ was observed in the northern forest steppe zone, followed by the central typical and western desert steppe zones, where grassland productivity was relatively low. Among the six eco-geographical regions, the average annual ANPP$_{NR}$ of the warm temperate semi-humid region reached a maximum value at 121.79 gCm$^{-2}$year$^{-1}$, while that of the medium-temperate arid region represented the lowest value with 82.11 gCm$^{-2}$year$^{-1}$.

The final ANPP$_{ES}$ in the Inner Mongolian grassland was determined using ANPP$_{SC}$, ANPP$_{SF}$, and ANPP$_{NR}$ together. Over the study period, ANPP$_{ES}$ ranged between 37.08 and 402.13 gCm$^{-2}$year$^{-1}$, with the regional average value of 115.37 gCm$^{-2}$year$^{-1}$ (Fig 2D). ANPP$_{ES}$ displayed decreasing gradients from the northeast to the southwest, with more than 80% of the area falling between 85 and 165 gCm$^{-2}$year$^{-1}$. Overall, the spatial distribution of ANPP$_{ES}$ was similar to that of ANPP$_{NR}$ because ANPP$_{NR}$ was higher than ANPP$_{SC}$ and ANPP$_{SF}$ in most areas. In the eastern part of the study area, under the effects of precipitation and topography, a higher ANPP was required for the SC than that for the SF and NR. In the central region, where drought and strong winds frequently occur, the demand for ANPP for SF was higher than that for SC and NR. In addition, the ANPP$_{ES}$ in different eco-geographical regions exhibited notable spatial heterogeneity. The warm-temperate-semi-humid region had the highest ANPP$_{ES}$ of 199.97 gCm$^{-2}$year$^{-1}$, while the medium-temperate arid region had the lowest ANPP$_{ES}$ of 98.82 gCm$^{-2}$year$^{-1}$.

## Spatial pattern of ECC

To ensure the sustainable use of various ecosystem services, the ECC of the Inner Mongolian grasslands from 1987–2015 was estimated (**Fig 3**). Theoretically, the total ECC of the Inner Mongolian grassland was 78.52 million sheep unit m$^{-2}$. The region with a negative ECC accounted for 4.18% of the total area and was concentrated in the east and northwest, indicating that the natural grassland in these areas had insufficient ability to supply provisioning services to humans. In areas with an ECC greater than zero, the average annual ECC was 1.59 sheep unit hm$^{-2}$. The spatial differentiation of ECC revealed that the ECC of the central region was higher than that of the eastern and western regions. For eco-regions, ECC varied across different zones, ranging from 0.77 to 1.69 sheep unit hm$^{-2}$. The medium-temperate semi-arid region had the highest ECC, followed by the medium-temperate humid region and medium-temperate sub-humid region, with average ECC of 1.30 and 1.34 sheep unit m$^{-2}$, respectively. Low levels of ECC were represented by the cold temperate humid region, medium-temperate arid region, and warm temperate semi-humid region, where the average ECC was less than 1.08 sheep unit m$^{-2}$ (**Fig 3. Spatial pattern of multi-year average ecological carrying capacity in Inner Mongolia grassland from 1987 to 2015.**).

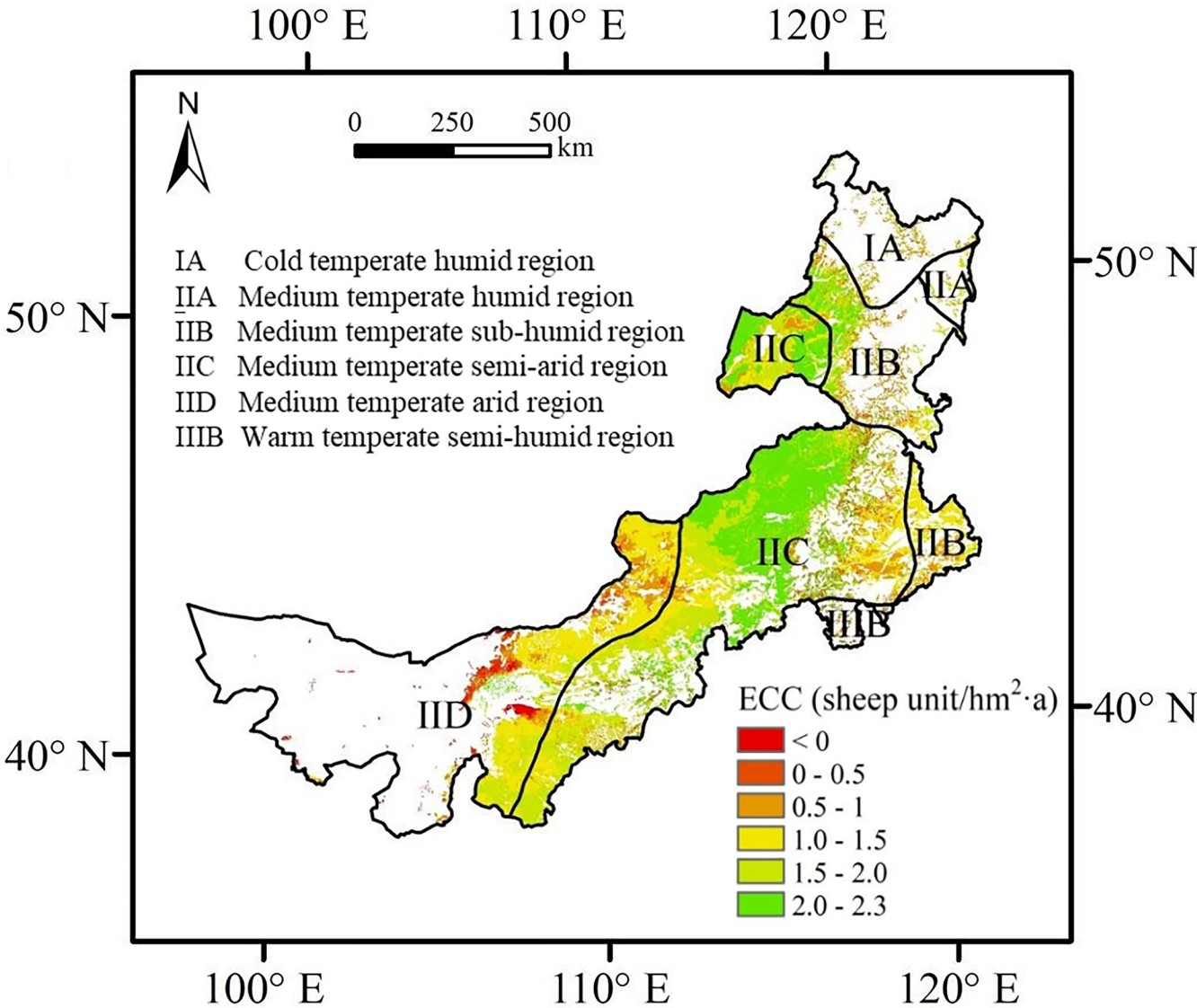

**Fig 3. Spatial pattern of multi-year average ecological carrying capacity in Inner Mongolia grassland from 1987 to 2015.**

### Driving factors of ECC

The impacts ($q$ values) of the eight influencing factors on the ECC of Inner Mongolian grasslands from 1987 to 2015 were obtained using a geographical detector. Eight factors were ranked in descending order by $q$ values as follows: NDVI (0.25) > precipitation (0.19) > soil type (0.17) > temperature (0.16) > sunshine duration (0.13) > slope (0.12) > wind speed (0.09) > elevation (0.06). NDVI had the greatest influence on the spatial heterogeneity of ECC in Inner Mongolia, followed by precipitation, soil type, and temperature, which were also important in determining the spatial pattern of ECC. The explanatory powers of the eight influencing factors varied considerably among the eco-geographical regions (**Fig 4**). In the cold temperate humid region, the slope had a much greater impact on the spatial pattern of ECC than the other factors, accounting for 19% of the ECC variation. In the medium-

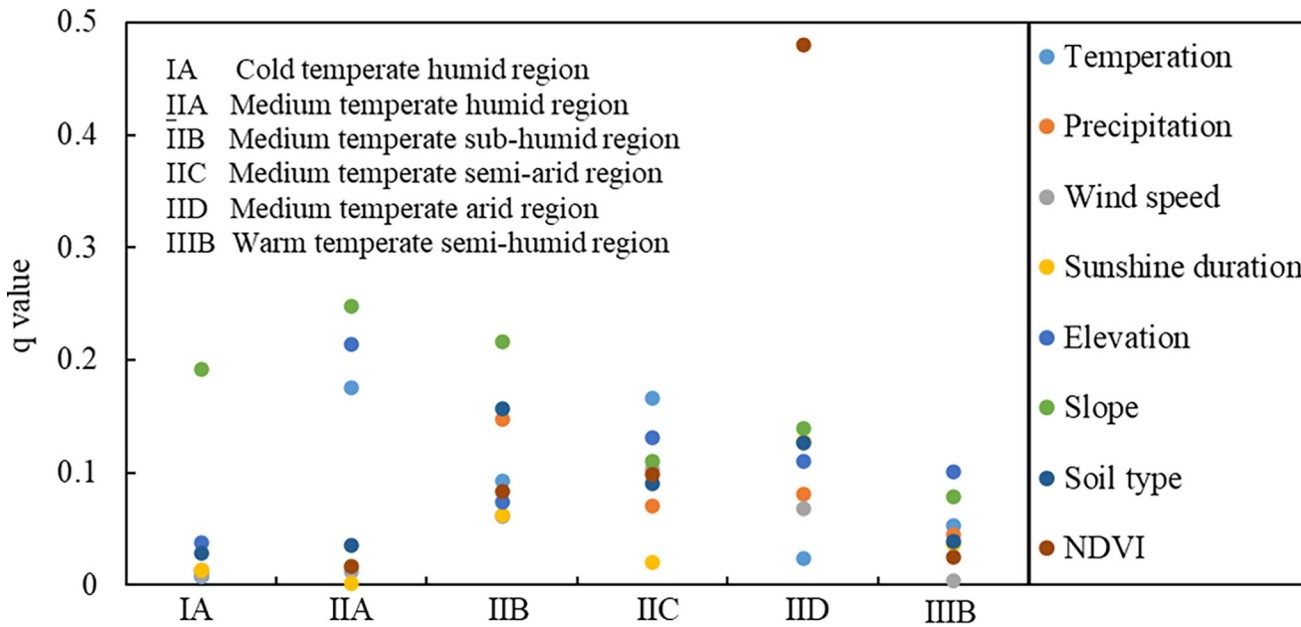

**Fig 4.** *q* values of influencing factors for different eco-geographical regions.

temperate humid region, the slope remained the dominant factor, followed by elevation and temperature. In the medium-temperate sub-humid region, the *q* value of the slope was higher than that of the other influencing factors, while soil type and precipitation also played vital roles. In the medium-temperate semi-arid region, temperature and elevation were the primary factors affecting the spatial distribution of ECC. The NDVI had a substantially higher explanatory power in the medium-temperate arid region, accounting for 48% of the ECC variation. In the warm temperate semi-humid region, the explanatory power of all the factors was relatively low, and elevation had the greatest impact (**Fig 4. *q* values of influencing factors for different eco-geographical regions.**).

The interactive results between influencing factors revealed that the interaction effects of paired factors on ECC distribution represented bivariate or nonlinear enhancement, indicating that the combinations of any pair of factors could play a more important role than their single effect. Specifically, the *q* value of the interaction between elevation and NDVI was the highest, followed by that between slope and NDVI. Furthermore, the dominant interaction affecting ECC differed among the eco-geographical regions. Table 2 summarizes the

**Table 2. The dominant interactions between two factors in different eco-geographical regions.**

| Eco-geographical region | Interaction 1 | Interaction 2 | Interaction 3 |
|---|---|---|---|
| IA | Elevation ∩ slope | Slope ∩ soil type | Precipitation ∩ slope |
| IIA | Elevation ∩ slope | Temperature ∩ slope | Slope ∩ soil type |
| IIB | Temperature ∩ slope | Slope ∩ NDVI | Slope ∩ soil type |
| IIC | Elevation ∩ NDVI | Temperature ∩ slope | Elevation ∩ slope |
| IID | Slope ∩ NDVI | Elevation ∩ NDVI | soil type ∩ NDVI |
| IIIB | Elevation ∩ slope | wind speed ∩ elevation | Elevation ∩ NDVI |

In IA, IIA, and IIB, the combinations of slope and other factors had a more dominant influence compared to all other combinations. In IIC, the interaction effect of elevation and NDVI had the greatest contribution to ECC, followed by the interaction effect of temperature and slope and elevation and slope. In IID, NDVI combined with other factors better explained the ECC distribution. In IIIB, the interactions involving elevation and other factors had a higher explanatory power.

interactions with the three highest explanatory powers. In the cold temperate humid region, medium-temperate humid region, and medium-temperate sub-humid region, the combinations of slope and other factors had more dominant influences among all the combinations. In the medium-temperate semi-arid region, the interaction effect of elevation and NDVI made the greatest contribution to ECC, followed by the interaction effect of temperature and slope and elevation and slope. NDVI combined with other factors could better explain the ECC distribution in the medium-temperate arid region. In the warm temperate semi-humid region, the interactions between elevation and other factors had a higher explanatory power.

## Discussion

### Comparison between ECC

ECC studies are necessary for identifying ways to achieve regional sustainable development, and some research efforts have been devoted to estimating ECC. For example, Zhang *et al.* simulated the temporal and spatial variation of grazing capacity in Inner Mongolia grassland from 1961–2010. The results revealed that proper grazing capacity was 0.76–1.60 sheep unit m$^{-2}$, consistent with our results [46]. Similarly, Surina *et al.* evaluated the theoretical livestock carrying capacity in Inner Mongolia based on the CASA model and discovered that the average livestock carrying capacity during 2011–2013 was 107.10 million sheep unit m$^{-2}$, which was higher than the total ECC evaluated in this study [47]. These efforts only roughly multiplied the grassland NPP/biomass and utilization rate to calculate the carrying capacity; however, they failed to consider that regulating services and the utilization rate require further verification. Recently, Dong *et al.* analyzed the carrying capacity and status of grasslands in the Mongolian Plateau. However, they focused on the ANPP supplied by the ecosystem while neglecting the trade-offs among ecosystem services and the prerequisites for ensuring the integrity of ecosystem services [48]. The framework developed in this study considers the sustainable use of various ecosystem services in the assessment of ECC, which has major implications for balancing socio-economic development and ecological protection.

### Mechanisms underlying the impacts of natural factors on ECC

Based on the results of the geographical detector, NDVI, precipitation, and soil type were the key determinants that considerably affected the distribution of ECC in Inner Mongolian grasslands. High ECC was mainly distributed in areas characterized by relatively good heat and moisture conditions, well-preserved vegetation, and flat terrain. Existing research explains the association between primary impact factors and ECC. NDVI is an important indicator of vegetation conditions and has been demonstrated to be strongly correlated with grassland ANPP [49]. Precipitation directly affects various biological processes and is the main driving force behind grassland ecosystem changes [50]. Soil types vary greatly regarding moisture content, soil structure, and nutrient supply, which directly affect soil erodibility, plant species, and grassland productivity [51]. ECC is associated with ecosystem biotic interactions; thus, the influencing factors do not work independently but interact together. When NDVI is coupled with other factors, the explanatory power of ECC distribution increases substantially [52].

### Implications for grassland ecosystem sustainable management

The comprehensive development of grassland ecosystems requires scientific development schedules based on grassland sustainability. Considering the conflicts among different ecosystem service usage, we should trade off socio-economic development and ecological conservation. Statistical data have revealed that the actual stocking rate in Inner Mongolia increased

from 70 million in 2001 to 105 million sheep units in 2016 [48], indicating that the supply of ecosystem services cannot meet human demand. In the east and northwest region with a negative ECC, the current AH production exceeds the ECC threshold, continued grassland exploitation will inevitably cause damage to ecosystems and substantially diminish the ecosystem services obtained from grasslands. Therefore, when designing regional strategies, decision-makers should effectively integrate the needs of pastoralists with environmental policies to promote the sustainable use of grasslands.

ECC hotspots are the priority areas for development in the medium-temperate semi-arid region with an ECC greater than zero. With a relatively stable ecosystem structure and adequate grassland resource supply, these areas have inevitably become priority development objects. The transportation infrastructure, meat processing and storage, and food safety precautions require improvements for the continued expansion of livestock markets [53].

However, the cold temperate humid region, medium-temperate arid region, and warm temperate semi-humid region with low levels of ECC have fragile ecological environments, simple ecosystems, and scarce potential resources, sustaining ecosystem services in ECC cold spots is difficult; thus, ecological conservation should be prioritized in these areas. Measures such as no grazing and light grazing are recommended, along with supplemental livestock feeding. Moreover, grassland policies and incentives are urgently needed to avoid further increases in stocking rates. Sustainable grassland management approaches that ensure synergy between ecosystems and socio-economic systems should be promoted.

## Limitations and uncertainty

Some uncertainty from the determination of parameters and thresholds is inevitable. In this study, the soil erosion threshold used to calculate $ANPP_{SC}$ and $ANPP_{SF}$ was set based on the widely used standards for the classification and gradation of soil issued by the Ministry of Water Resources of the People's Republic of China. However, the criteria for determining soil loss tolerance have certain limitations. Duan *et al.* conducted a field investigation to calculate the soil loss tolerance of 21 soil species in Northeast China. They discovered that the average value was 29.5% lower than the current national standard [54]. Therefore, the determination of soil loss tolerance requires further improvement. In addition, the ANPP required for NR was specified as 50% of the ANPP. The optimal grassland utilization rates proposed by different scholars are inconsistent; therefore, more studies are required to determine the threshold of $ANPP_{NR}$ in different regions.

ANPP is a good indicator of ECC; however, not all NPP of grassland types and vegetation species can be used for grazing activities. For example, poisonous species sometimes have a larger NPP than edible forage but cannot be used by yaks or sheep. Similarly, shrubs cannot be used as efficiently as grass [55].

In addition, based on the characteristics of grassland ecosystems, we selected SC, SF, and NR as the key ecosystem services for evaluation. However, grasslands also provide other eco-system services, such as water regulation and carbon sequestration; thus, the ECC in Inner Mongolian grasslands may have been overestimated [56]. More targeted ecosystem services should be selected to suit each case when the research framework is applied to other regions. More ecosystem services and functions can be added according to the types and functions of local ecosystems in other vulnerable eco-regions. However, calculating the fraction of ANPP required for other ecosystem services is difficult and should be explored further. In addition, the uneven distribution of meteorological stations could inevitably increase the uncertainty in vulnerable eco-regions.

## Conclusions

ECC is regarded as the key to guiding regional ecosystem service trade-offs and achieving sustainability objectives in vulnerable eco-regions. We developed a framework for ECC evaluation focusing on the allocation of ANPP between AH and the maintenance of various ecosystem services, providing a valuable threshold for policymakers to identify the sustainable development potential from the perspective of ecosystem services in vulnerable eco-regions. Using this framework, we evaluated the ECC of the Inner Mongolian grasslands. The total ECC of Inner Mongolian grassland was 78.52 million sheep units/a. The average annual ECC was 1.59 sheep unit $hm^{-2}$ from 1987–2015, which revealed gradients increasing from the southwest to the northeast. NDVI, precipitation, and soil type were the primary factors affecting the ECC distribution in the Inner Mongolian grasslands. Statistical data have revealed that the actual stocking rate in Inner Mongolia increased from 70 million to 105 million sheep units, indicating that current AH production has exceeded the ECC threshold. Continued grassland exploitation inevitably damages the grassland ecosystems.

Maintaining the sustainability of ecosystem services in vulnerable eco-regions with fragile ecological environments, simple ecosystems, or scarce potential resources is difficult. Measures such as no grazing and light grazing are recommended, along with supplemental livestock feeding. Moreover, grassland policies and incentives are needed to avoid further increases in stocking rates.

## Supporting information

**S1 Raw images.**
(DOCX)

**S1 Table. A list of acronyms and the supplementary methods description.**
(DOCX)

**S1 File.**
(ZIP)

## Acknowledgments

We would like to thank Editage (www.editage.com) for English language editing. We thank Dongsheng-Zhao for providing us with the research method for calculating the ecological carrying capacity. We also thank the anonymous reviewers and editors for their valuable, constructive comments.

## Author Contributions

**Conceptualization:** Caiyun Guo.

**Data curation:** Shuyu Song, Gege Xie.

**Formal analysis:** Caiyun Guo.

**Investigation:** Caiyun Guo, Shilin Luo.

**Methodology:** Caiyun Guo, Dongsheng Zhao.

**Project administration:** Dongsheng Zhao.

**Resources:** Caiyun Guo.

**Software:** Shuyu Song.

**Supervision:** Dongsheng Zhao, Lingchun Yang.

**Visualization:** Shuyu Song.

**Writing – original draft:** Shuyu Song.

**Writing – review & editing:** Caiyun Guo.

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
