## [Decision Letter · Decision Letter 0]

2 May 2023

PONE-D-23-10372Quantifying the ecological carrying capacity of grasslands in Inner MongoliaPLOS ONE

Dear Dr. Guo,

Thank you for submitting your manuscript to PLOS ONE. After careful consideration, we feel that it has merit but does not fully meet PLOS ONE’s publication criteria as it currently stands. Therefore, we invite you to submit a revised version of the manuscript that addresses the points raised during the review process.

We look forward to receiving your revised manuscript.

Kind regards,

Chun Liu

Academic Editor

PLOS ONE

Journal Requirements:

   "Scientific research Project of Education Department of Hunan Province (21C0744) "

6. We note that Figures 1-3 in your submission contain [map/satellite] images which may be copyrighted. All PLOS content is published under the Creative Commons Attribution License (CC BY 4.0), which means that the manuscript, images, and Supporting Information files will be freely available online, and any third party is permitted to access, download, copy, distribute, and use these materials in any way, even commercially, with proper attribution. For these reasons, we cannot publish previously copyrighted maps or satellite images created using proprietary data, such as Google software (Google Maps, Street View, and Earth). For more information, see our copyright guidelines: http://journals.plos.org/plosone/s/licenses-and-copyright.

a. You may seek permission from the original copyright holder of Figures 1-3 to publish the content specifically under the CC BY 4.0 license.  

7. Please include a copy of Table 1 and 2 which you refer to in your text on pages 19 and 24.

Additional Editor Comments:

after reading the paper, the author should strengthen the statement of research novelty. meanwhile, the result and discussion analysis should be deepened. please revise the ms thoroughly.

Reviewers' comments:

Reviewer's Responses to Questions

**Comments to the Author**

1. Is the manuscript technically sound, and do the data support the conclusions?

Reviewer #1: Yes

2. Has the statistical analysis been performed appropriately and rigorously? 

Reviewer #1: Yes

3. Have the authors made all data underlying the findings in their manuscript fully available?

Reviewer #1: Yes

4. Is the manuscript presented in an intelligible fashion and written in standard English?

Reviewer #1: Yes

5. Review Comments to the Author

Reviewer #1: This study presents a calculation framework for determining ecological carrying capacity in vulnerable areas using data from animal husbandry, soil conservation, sand fixation, and natural regeneration. The framework was applied to assess the ecological carrying capacity of Inner Mongolia (measured in sheep unit hm-2) and identify drivers of spatial heterogeneity. While the topic is interesting, the writing could benefit from further improvement. Therefore, I suggest a minor revision.

The introduction does not adequately support the novelty and importance of this study. Additional studies should be cited to demonstrate that previous research has overlooked correlations between environmental and social factors.

The authors aim to propose an improved calculation framework for ecological carrying capacity, while they do not provide sufficient evidence supporting its advantages over existing methods. Comparing their results with those of other studies on spatial distribution would strengthen their argument.

Many abbreviations are unnecessary, such as ES, MA, IA IID and IIIB, which actually increase the difficulty of reading.

Line 175-176: The formula layout is incorrect and needs further adjustment.

Figure 4: The image needs to be self-explanatory. If abbreviations are used, they need to be explained in the Figure caption

6. PLOS authors have the option to publish the peer review history of their article (what does this mean?). If published, this will include your full peer review and any attached files.

Reviewer #1: No

---

## [Author Response · Author response to Decision Letter 0]

9 Aug 2023

Dear editor Liu and reviewers: 

We would like to express our great appreciation to you for your efforts in the previous version of the manuscript. You gave very valuable suggestions and comments for the paper which are of great help for us to improve our manuscript. We have studied reviewer’s comments carefully. According to the reviewer’s detailed suggestions, we have made a careful revision on the original manuscript. All revised portions are marked using track changes. Here we would like to submit the revised version comply with the PLOS ONE submission guidelines. We hope that the revised manuscript is satisfactory to your journal.

Journal Requirements:

1. Please ensure that your manuscript meets PLOS ONE's style requirements, including those for file naming. The PLOS ONE style templates can be found at https://journals.plos.org/plosone/s/file?id=wjVg/PLOSOne_formatting_sample_main_body.pdf and https://journals.plos.org/plosone/s/file?id=ba62/PLOSOne_formatting_sample_title_authors_affiliations.pdf.

Response: Thank you for your comments. We have modified the manuscript as the PLOS ONE style templates (Details are in the “Revised Manuscript with Track Changes”).

● The name of the colleague or the details of the professional service that edited your manuscript.

● A copy of your manuscript showing your changes by either highlighting them or using track changes (uploaded as a *supporting information* file).

● A clean copy of the edited manuscript (uploaded as the new *manuscript* file).

Response: Thank you very much for your professional recommendation. The manuscript has been edited by professional editors Vipin at Editage to ensure language and grammar accuracy. The logical presentation of ideas and the structure of the paper were also checked during the editing process. The author's core research ideas were not altered during the editing process.

 Response: I apologize for the oversight in not sufficiently including the permission details for the AGB sampling. Your gentle reminder is valued. In the Methods section, we obtained the AGB data from a field survey carried out in Inner Mongolia at the peak of grassland biomass. Given that our surveys were conducted in spacious, unobtrusive grasslands with negligible environmental impact, no permits were necessary.

 "Scientific research Project of Education Department of Hunan Province (21C0744) "

Response: Thank you for your comments. We have stated the funders and its role in this study in our cover letter: “This study was supported by the Scientific Research Foundation of the Hunan Provincial Education Department (21C0744). The funder had no role in study design, data collection and analysis, the decision to publish, or the preparation of the manuscript.” (Line 36-38).

"Upon re-submitting your revised manuscript, please upload your study minimal underlying data set as either Supporting Information files or to a stable, public repository and include the relevant URLs, DOIs, or accession numbers within your revised cover letter. For a list of acceptable repositories, please see http://journals.plos.org/plosone/s/data-availability#loc-recommended-repositories. Any potentially identifying patient information must be fully anonymized. 

Response: Thank you for your comments. We have uploaded the data as “supporting information”.

6. We note that Figures 1-3 in your submission contain [map/satellite] images which may be copyrighted. All PLOS content is published under the Creative Commons Attribution License (CC BY 4.0), which means that the manuscript, images, and Supporting Information files will be freely available online, and any third party is permitted to access, download, copy, distribute, and use these materials in any way, even commercially, with proper attribution. For these reasons, we cannot publish previously copyrighted maps or satellite images created using proprietary data, such as Google software (Google Maps, Street View, and Earth). For more information, see our copyright guidelines: http://journals.plos.org/plosone/s/licenses-and-copyright. 

 a. You may seek permission from the original copyright holder of Figures 1-3 to publish the content specifically under the CC BY 4.0 license. 

We recommend that you contact the original copyright holder with the Content Permission Form (http://journals.plos.org/plosone/s/file?id=7c09/content-permission-form.pdf) and the following text: “I request permission for the open-access journal PLOS ONE to publish XXX under the Creative Commons Attribution License (CCAL) CC BY 4.0 (http://creativecommons.org/licenses/by/4.0/). Please be aware that this license allows unrestricted use and distribution, even commercially, by third parties. Please reply and provide explicit written permission to publish XXX under a CC BY license and complete the attached form.” Please upload the completed Content Permission Form or other proof of granted permissions as an "Other" file with your submission. In the figure caption of the copyrighted figure, please include the following text: “Reprinted from [ref] under a CC BY license, with permission from [name of publisher], original copyright [original copyright year].” b. If you are unable to obtain permission from the original copyright holder to publish these figures under the CC BY 4.0 license or if the copyright holder’s requirements are incompatible with the CC BY 4.0 license, please either i) remove the figure or ii) supply a replacement figure that complies with the CC BY 4.0 license. Please check copyright information on all replacement figures and update the figure caption with source information. If applicable, please specify in the figure caption text when a figure is similar but not identical to the original image and is therefore for illustrative purposes only. The following resources for replacing copyrighted map figures may be helpful: 

USGS National Map Viewer (public domain): http://viewer.nationalmap.gov/viewer/The Gateway to Astronaut Photography of Earth (public domain): http://eol.jsc.nasa.gov/sseop/clickmap/Maps at the CIA (public domain): https://www.cia.gov/library/publications/the-world-factbook/index.html and https://www.cia.gov/library/publications/cia-maps-publications/index.html NASA Earth Observatory (public domain): http://earthobservatory.nasa.gov/Landsat: http://landsat.visibleearth.nasa.gov/USGS EROS (Earth Resources Observatory and Science (EROS) Center) (public domain): http://eros.usgs.gov/#Natural Earth (public domain): http://www.naturalearthdata.com/.

Response: Thank you for your comments. The images were obtained by using ArcGIS 10.2 through the open-access data process. The spatial extent of the Inner Mongolia was obtained from the Resource and Environmental Science Data Center of the Chinese Academy of Sciences (http://www.resdc.cn/). They are all the open-access data.

7. Please include a copy of Table 1 and 2 which you refer to in your text on pages 19 and 24.

Response: Thank you for your comments. We have added Tables 1 and 2 mentioned in the text to the manuscript (Line 262-263, 409-410).

Additional Editor Comments:

after reading the paper, the author should strengthen the statement of research novelty. meanwhile, the result and discussion analysis should be deepened. please revise the ms thoroughly.

Response: Thank you for your comments. We have included captions for our Supporting Information files at the end of the manuscript, and update the citations to match accordingly (Line 728-731).

We strengthen the statement of research novelty in the Introduction section (Line 134-157). And added the “Implications for grassland ecosystem sustainable management” section to analysis the research deeply (Line 452-476). 

We have revised the manuscript thoroughly according the reviewer’s requirements. 

Comments to the Author

Reviewer #1: This study presents a calculation framework for determining ecological carrying capacity in vulnerable areas using data from animal husbandry, soil conservation, sand fixation, and natural regeneration. The framework was applied to assess the ecological carrying capacity of Inner Mongolia (measured in sheep unit hm-2) and identify drivers of spatial heterogeneity. While the topic is interesting, the writing could benefit from further improvement. Therefore, I suggest a minor revision.

The introduction does not adequately support the novelty and importance of this study. Additional studies should be cited to demonstrate that previous research has overlooked correlations between environmental and social factors. 

Response: Thank you for your comments. We cited the two papers to illustrate the importance of the harmonious coexistence of the natural environment and societal development.

 [2] Ostad-Ali-Askari, K., Shayannejad, M., 2020. Impermanent changes investigation of shape factors of the volumetric balance model for water development in surface irrigation. Modeling Earth Systems and Environment. 6, 1573–1580.

[3] Nafchi, R.F., Samadi-Boroujeni, H., Vanani, H.R., Ostad-Ali-Askari, K., Brojeni, M.K., 2021. Laboratory investigation on erosion threshold shear stress of cohesive sediment in Karkheh Dam. Environ. Earth Sci. 80, 19.(Line 68-69).

We cited the paper of Ostad-Ali-Askari and Shayannejad (2021) to illustrate the importance of calculating the ECC :

[9] Ostad-Ali-Askari, K., Shayannejad, M., 2021a. Computation of subsurface drain spacing in the unsteady conditions using Artificial Neural Networks (ANN). Appl. Water Sci. 11 (2), 21 (Line 77).

We cited the three papers to demonstrate that previous research has overlooked correlations between environmental and social factors. 

[15] Wu XL, Hu F. Analysis of ecological carrying capacity using a fuzzy comprehensive evaluation method. Ecological Indicators. 2020; 113: 106243. https://doi.org/10.1016/j.ecolind.2020.106243 . 

[16] Bu J, Li, C, Wang, X, Wang X, Zhang Y, Yang ZW. Assessment and prediction of the water ecological carrying capacity in Changzhou city, China. Journal of Cleaner Production. 2020; 277: 123-988. https://doi.org/10.1016/j.jclepro.2020.123988 .

[17] Thiesen T, Bhat MG, Liu H, Rovira R. An ecosystem service approach to assessing agro-ecosystems in urban landscapes. Land. 2022; 11 (4), 469. https://doi.org/10.3390/land11040469 (Line 88-93).

The authors aim to propose an improved calculation framework for ecological carrying capacity, while they do not provide sufficient evidence supporting its advantages over existing methods. Comparing their results with those of other studies on spatial distribution would strengthen their argument. 

Response: Thank you for your comments. There are relatively few literatures on the calculation of ecological carrying capacity of Inner Mongolia grassland. We have compared our result with other studies on spatial distribution in the “Comparison between ECC” section (Line 418-436).

Many abbreviations are unnecessary, such as ES, MA, IA IID and IIIB, which actually increase the difficulty of reading.

Response: Thank you for your comments. We have removed unnecessary abbreviations, such as ES, MA, IA, IIA, IIB, IIC, IID and IIIB. 

Line 175-176: The formula layout is incorrect and needs further adjustment.

Response: Thank you for your comments. We have adjusted the formula layout (Line 208, 222-223).

Figure 4: The image needs to be self-explanatory. If abbreviations are used, they need to be explained in the Figure caption.

Response: Thank you for your comments. We have explained the abbreviations in the each Figure caption.

6. PLOS authors have the option to publish the peer review history of their article (what does this mean?). If published, this will include your full peer review and any attached files.

Do you want your identity to be public for this peer review? For information about this choice, including consent withdrawal, please see our Privacy Policy.

Reviewer #1: No

Response: Thank you for your comments. Our choice is No.

Response: Thank you for your comments. We have checked the figures to ensure they meet PLOS requirements.

Once again, thank you very much for your constructive comments and suggestions which would help us both in English and in depth to improve the quality of the paper.

Kind regards,

authors

E-mail: guocy.17b@igsnrr.ac.cn

---

## [Decision Letter · Decision Letter 1]

29 Aug 2023

Quantifying the ecological carrying capacity of grasslands in Inner Mongolia

PONE-D-23-10372R1

Dear Dr. Guo,

We’re pleased to inform you that your manuscript has been judged scientifically suitable for publication and will be formally accepted for publication once it meets all outstanding technical requirements.

Kind regards,

Chun Liu

Academic Editor

PLOS ONE

Additional Editor Comments (optional):

Reviewers' comments:

Reviewer's Responses to Questions

**Comments to the Author**

1. If the authors have adequately addressed your comments raised in a previous round of review and you feel that this manuscript is now acceptable for publication, you may indicate that here to bypass the “Comments to the Author” section, enter your conflict of interest statement in the “Confidential to Editor” section, and submit your "Accept" recommendation.

Reviewer #1: All comments have been addressed

2. Is the manuscript technically sound, and do the data support the conclusions?

Reviewer #1: Yes

3. Has the statistical analysis been performed appropriately and rigorously? 

Reviewer #1: Yes

4. Have the authors made all data underlying the findings in their manuscript fully available?

Reviewer #1: Yes

5. Is the manuscript presented in an intelligible fashion and written in standard English?

Reviewer #1: Yes

6. Review Comments to the Author

Reviewer #1: The manuscript has been well improved.

Suggestions for modifications:

Results and Discussion section: It is recommended to refine the policy recommendations proposed, list areas with high and low ECC values, and provide more specific or innovative policy recommendations for sub-regions.

Line 92-132: Please streamline the unnecessary use of filler words, such as “however,” “therefore,” and “hence,” and so on. Modify all instances of such usage, as removing these words does not significantly affect the meaning of the sentences.

Line 174, 310, 318, 322, 341, 356: “Fig 2a” � “Fig. 2a”. Please review all similar expressions in the article and revise them.

Line 408: “Elevation ∩ slope” � “Elevation ∩ Slope”. Please revise the table.

7. PLOS authors have the option to publish the peer review history of their article (what does this mean?). If published, this will include your full peer review and any attached files.

Reviewer #1: **Yes: **Ronghai HU

---

## [Editor Report · Acceptance letter]

10 Nov 2023

PONE-D-23-10372R1 

Quantifying the ecological carrying capacity of grasslands in Inner Mongolia 

Dear Dr. Guo:

I'm pleased to inform you that your manuscript has been deemed suitable for publication in PLOS ONE. Congratulations! Your manuscript is now with our production department. 

Kind regards, 

on behalf of

Dr. Chun Liu 

Academic Editor

PLOS ONE